# Is Resilience a Trait or a Result of Parental Involvement? The Results of a Systematic Literature Review

**Karolina Eszter Kovács** [1], **Beáta Dan** [2], **Anett Hrabéczy** [3], **Katinka Bacskai** [3,*] and **Gabriella Pusztai** [3]

1 Institute of Psychology, University of Debrecen, H-4032 Debrecen, Hungary; kovacs.karolina@arts.unideb.hu
2 Bonitas Special Education Center, 410032 Oradea, Romania; dan.beata@yahoo.com
3 Institute of Educational Studies and Cultural Management, University of Debrecen, H-4032 Debrecen, Hungary; hrabeczy.anett@arts.unideb.hu (A.H.); pusztai.gabriella@arts.unideb.hu (G.P.)
* Correspondence: bacskai.katinka@arts.unideb.hu; Tel.: +36-30-678-3055

**Abstract:** Investigating parental involvement has moved to the foreground of research in the past two decades, and research results focusing on family engagement claim its positive impact on children's academic and non-academic achievement. However, less is known about parental involvement in the case of families with children with special needs. In our systematic review, we collected studies focusing on parental involvement which emphasised the role of resilience. Using the EBSCO Discovery Service, a total of 467 abstracts from 85 databases were screened, of which 28 papers published between 1984 and 2021 met the research criteria. Papers vary according to methodology (interview, focus group conversation, survey, case study, intervention programme and good practice) and disability group (general or specific). Resilience is interpreted in two ways: as a personality trait or a consequence. Four types of papers could be detected which dealt with the target group, specifically papers focusing on children, parents, teachers and professionals, and intervention programmes with multiple focuses. In conclusion, resilience is an element of parental involvement, either as a personality trait or a result. It is indispensable for the successful development of children in terms of academic and non-academic achievement as well. Programmes providing a wider collaboration with actors involved in the development of children seem to be more effective. In general practice, whether the goal is to build upon resilience as a personality trait or target its development as a consequence, strong collaboration between the parents, teachers and professionals concerned in the process can significantly contribute to the child's psychological, emotional and academic development.

**Keywords:** parental involvement; children with special needs; resilience

## 1. Introduction

In recent decades, research has focused on how to most effectively educate children with special educational needs [1–3]. Most research interprets special education and inclusive education as competing paradigms, several aspects of which are discussed [3]. The public education system aims to provide the best for most children; however, for students who do not belong to this majority, the opportunities provided by the system are not always appropriate [4,5]. They often struggle to perform well in the majority of schools because they are more likely to face learning or behavioural problems. How this can be changed is still the focus of research, and its potential is being explored from several perspectives.

In this analysis, we are interested in how the school involvement of one of the most important actors in children's lives, the parent, appears in research and the results of previous research. In this study, we focus primarily on research that focuses on the engagement of parents of children with special educational needs who are educated inclusively. We see this as particularly important because parental involvement may be more problematic for these families, and it is likely that parents raising children with special educational needs will

need more support in working with professionals and engaging in their children's studies. The exploration of the topic is also justified by the fact that it is not only research which deals with the cooperation between professionals and parents of children with special educational needs, but in several countries, education policy also focuses on the effective implementation of these partnerships, mostly since the Warnock Report [6–9].

### 1.1. Definition of Special Educational Needs in Various Contexts

Defining the phenomenon is an important but very sensitive issue when examining social groups which deviate from the norm. In the various fields of school and education, the concept of special educational needs is most often encountered in terms of the fulfilment of study requirements. However, these concepts, especially the interpretation of disability, can be described by different definitions according to different disciplines. Thus, we may encounter different interpretations in health care and education, and sometimes, in addition to medical, psychological and sociological aspects, we may also encounter different interpretations in terms of the legal aspects of the issue [10,11]. It can be stated that the common denominator in the definitions used in educational research is based on the educational policy approach. This concept describes, within a school setting, students for whom each country makes additional resources available to enable these students to access the curriculum and progress effectively in their studies (OECD 2004). However, the interpretation of disability and special educational needs, and the additional services provided by institutions, varies greatly from country to country.

### 1.2. Concept of Special Educational Needs in the Analysis

Special educational needs mean different things from country to country, so comparisons are very difficult when looking at research results internationally. To address this, we used the OECD's cross-national categorisation, which the organisation created to make individual countries comparable and make the issue of special educational needs understandable in an international context [12]. The OECD created the following categories in 2004: SEN-A includes disabilities and impairments. According to the OECD [12], the special educational need for SEN-A students is primarily due to disability. The SEN-B category includes difficulties, typically behavioural or emotional disorders or problems, which present learning difficulties. The special educational need for these students can be traced back to problems in the interaction between the educational context and the student [12]. The SEN-C category includes disadvantages. For these students, special needs arise from cultural, socioeconomic or linguistic difficulties, and the aim for these students is to compensate for the resulting disadvantages. As the difficulties of SEN-C are typically due to different types of problems (mostly related to environmental disadvantage) compared to SEN-A and B, research on SEN-C students was excluded from the analysis.

However, it is important to note that this category system only provides a framework for understanding and comparing how each country interprets the SEN; on many points, it does not unify exactly which problems fall into each category. In its 2004 publication, the OECD systematises which problems each country classifies. In order to define the framework of the analysis, we standardised this systematisation for the SEN-A and SEN-B categories, the summary of which is illustrated in Table 1. The reason for standardisation is that a given disorder is not classified in the same SEN category by each country. In these cases, we made a researcher decision and, during standardisation, placed it in the category in which it is ranked by most countries.

**Table 1.** Categorisation of special needs based on the OECD's cross-national categorisation (2004) [12].

| Category | SEN-A/Disabilities | SEN-B/Difficulties |
|---|---|---|
| Description | Disabilities or impairments, sensory, motor or neurological defects | Behavioural or emotional disorders, specific difficulties in learning |
| Special needs | Arise from problems connected to these disabilities | Arise from the interaction between the educational context and the student |
| Types | Intellectual disabilities, fragile X syndrome, Down syndrome, developmental delay, PWS, visual impairments, hearing impairments, physical disabilities and mobility impairments, multiple handicaps | Learning disabilities and learning disorders, speech disorders, autism spectrum disorder, ADD, ADHD, behavioural disorders and difficulties |

*1.3. Parental Involvement and Resilience*

Previous research results highlight that parental involvement is a significant predictor of children's academic and non-academic achievement [13,14]. From the school side, teachers have a crucial role to play in promoting family engagement by supporting the involvement of families in education in the home setting and by fostering a collaborative partnership between family and school [15]. These statements are generally true for children attending school. However, the family dynamics of families with children with special education needs usually differ from those of average families. These families have to adapt to the needs of the children, which often leads to multiple tasks and changed focus. Therefore, it is also well-known that children and young people with disabilities and their families often face adversities and challenges throughout their lifespan, which require coping skills, flexibility and a resilient personality. Resilience is a universal capacity that enables an individual, a group or a community to prevent, minimise or overcome the negative effects of adversity [16,17]. It can support academic achievement [18] and non-academic achievement, e.g., health behaviour [19]. Since resilience integrates intrapersonal strength, interventions also often build upon the observation of strengths [20], where the focus is not on the use of strengths themselves, but on the individual's motivation and ability to discover strengths in others. The observation of strengths can thus be defined as the ability to identify and observe one's own and others' strengths [21,22], contributing to resilience.

In our research, we aimed to investigate the parental involvement of families with children with special needs from the perspective of resilience.

## 2. Methods

This systematic literature review follows the Preferred Reporting Items for Systematic Reviews and Meta-Analyses (PRISMA) guidelines [23]. (See Figure 1.)

*2.1. Eligibility Criteria*

This systematic review met the following inclusion criteria: (1) reported original, empirical research published in a peer-reviewed journal, (2) which examined the relationship between families with children with special educational needs and the school, (3) was published in English, and (4) was in the following disciplines: education, psychology, social work, sociology, social sciences and humanities. This study did not examine kindergartens and non-empirical studies. We examined only journal articles. Books/book chapters, dissertations and newspaper articles were excluded.

*2.2. Search Strategy*

We consulted a research librarian from the University of Debrecen about our searching strategy. We performed searches on 20 October 2021 in the EBSCO Discovery Service search engine, which includes 85 databases (for information sources, see Appendix A). We used

general keywords to select the literature formulated in the first phase of the review study. These general keywords were:

1. parent involvement OR parent participation OR parent engagement OR family involvement OR family engagement;
2. parents of children with disabilities OR family with children with disability internalizing problems OR developmental problems OR conduct disorder OR children with disabilities OR inclusive education OR special education OR special need OR disabilities OR visual impairments OR hearing impairments OR intellectual disabilities OR cognitive impairments OR handicap OR speech impairments OR deaf OR blind OR autism OR physical disability OR visual disability OR mental disorder OR hearing disability OR visual disability OR disorder OR disabled;
3. resilience OR resiliency OR resilient OR strengths OR coping OR hardiness OR adaptation.

Overall, our systematic searches yielded 442 records, and after the double filtering, we excluded 137 records. We additionally searched in 11 specialist journals. We searched for our keywords in the magazines' search engine, during which we added 183 unique results to our list.

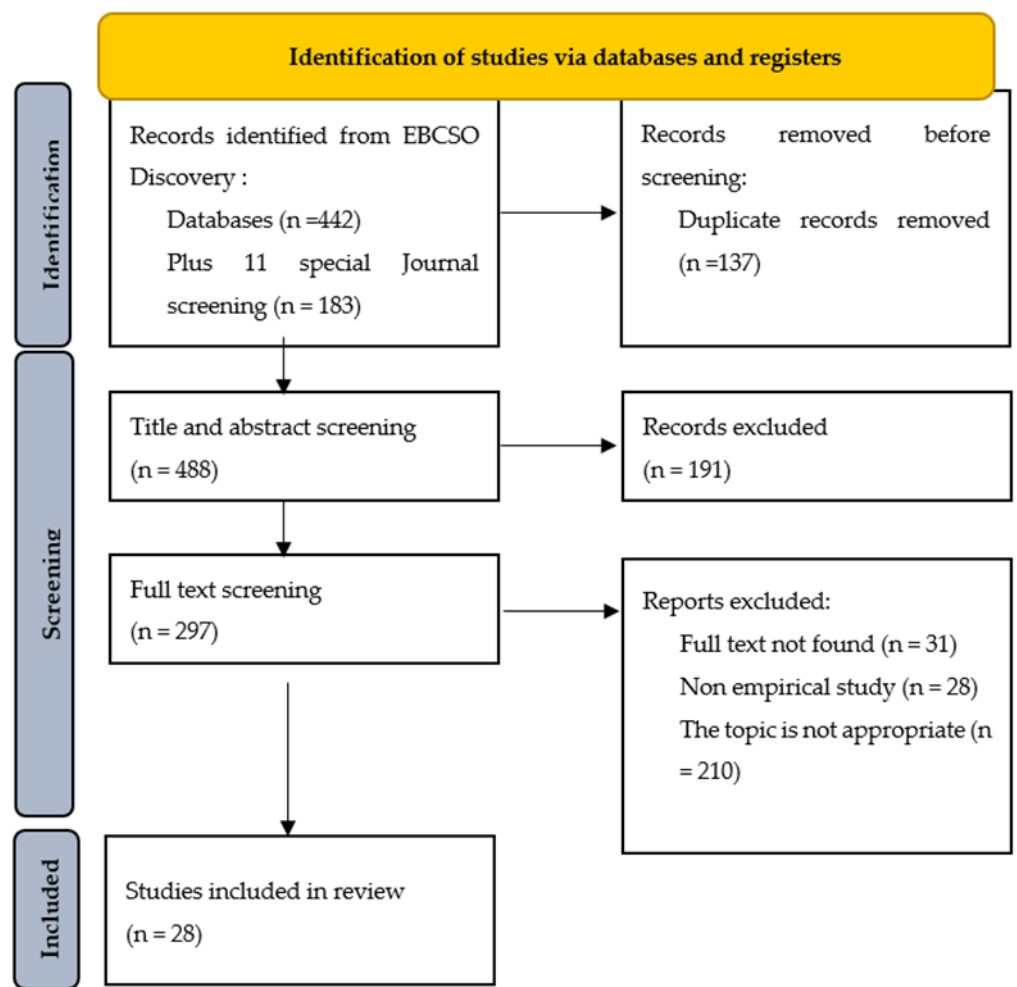

**Figure 1.** Preferred Reporting Items for Systematic Reviews and Meta-Analyses (PRISMA) diagram.

*2.3. Study Selection*

After the removal of duplicate studies, we performed a multistage screening process to select those studies which met the inclusion criteria:

- Stage 1, screening of titles and abstracts: the first review author screened the titles and abstracts of all identified records (KK). Twenty-five per cent of all titles and

abstracts were independently assessed by a second review author (HA, DB, BK). All studies whose adequacy was questionable were taken forward to the full-text screening at this stage.

- Stage 2, screening of full texts: two review authors (KK, HA, DB, BK) independently screened all full texts. In cases of uncertainty, the other authors also checked the decision.

## 3. Results

Overall, our systematic searches yielded 488 unique records. After title and abstract screening, 279 records were carried forward for full-text screening (Figure 1). Of these, 269 full-text papers were assessed for eligibility. Twenty-eight of them were non-empirical studies (reviews and meta-analyses). The full texts of 31 articles could not be retrieved. The characteristics of the studies involved in the review see in Table 2.

**Table 2.** The characteristics of the studies involved in the review.

| Type of Disability | | Authors | Date | Research Methods | Target Group | Region |
|---|---|---|---|---|---|---|
| General | | Santamaría Graff et al. | 2021 | intervention, focus group | professionals | USA |
| General | | Lavan et al. | 2019 | survey | parents | Israel |
| General | | Apodaca et al. | 2015 | good practice, survey | children, parents, teachers | USA |
| General | | Buchner et al. | 2014 | interview | children | Austria, Czech Republic, Ireland, Spain |
| General | | Flynn et al. | 2013 | survey | children | Canada |
| General | | Stanley | 2015 | interview | parents | USA |
| General | | Linder and Garnett | 1984 | survey | parents | USA |
| General | | Blackman and Mahon | 2016 | case study | parents, teachers, school | Barbados |
| General | | Swoszowski and Rollins | 2019 | intervention, case study | parents, teachers, professionals, school | USA |
| General | | Rodriguez et al. | 2014 | focus group | parents | USA |
| General | | Gedfie et al. | 2021 | survey | parents | Ethiopia |
| General | | Singal et al. | 2021 | interview | parents, teachers | Malawi, England, Sweden |
| General | | Slowík et al. | 2021 | intervention | professionals, school | Czech Republic |
| General | | Ravet and Mtika | 2021 | survey | teachers | Cambodia |
| General | | Timothy and Agbenyega | 2019 | interview, focus group | teachers | Australia |
| General | | Lendrum et al. | 2015 | survey, interview | children, parents, school | England |
| Specific | ADD/ADHD | Morris et al. | 2019 | survey | children, parents, teachers, professionals, school | USA |
| Specific | ADD/ADHD | Power et al. | 2009 | good practice, case study | parents, teachers | USA |

**Table 2.** *Cont.*

| Type of Disability | | Authors | Date | Research Methods | Target Group | Region |
|---|---|---|---|---|---|---|
| Specific | Hearing disability | Smith and Prelock | 2002 | intervention, case study | children, parents, teachers, professionals, school | USA |
| Specific | ADD/ADHD | Fabiano et al. | 2021 | intervention | children, parents, teachers | USA |
| Specific | Autism | Shochet et al. | 2019 | intervention, interview | parents | Australia |
| Specific | Autism | Casillas et al. | 2017 | interview | parents | USA |
| Specific | Learning disability | Pentyliuk | 2002 | interview | parents | Canada |
| Specific | Learning disability | Phillips et al. | 2000 | case study | children | USA |
| Specific | Emotional/behavioural disorder | Carlson et al. | 2020 | interview | parents | USA |
| Specific | Communication disorder | Miron | 2010 | interview | parents | USA |
| Specific | Intellectual disability | Stainton and Besser | 1998 | interview | parents | Canada |
| Specific | Intellectual disability | Karisa et al. | 2021 | interview | children, parents, teachers | Kenya |

Overall, 28 papers met the criteria (see Appendix B). The papers were published between 1984 and 2021. However, most of them were published after 2000 (40), especially in the 2010s (22). Since there were only two papers which focused on international differences, most of the papers introduced their research results in light of a national or regional study. Most research was conducted in the United States (14).

According to the types of disabilities, we could detect two main groups of studies, including general studies introducing parental involvement in a general manner (without differentiating the various disabilities) and specific studies examining parental involvement in light of a particular disorder. This group could be divided into two subgroups based on the OECD SENDDDD categorisation: articles introducing parental involvement in light of disorders belonging to the SEN-A category (hearing disability, intellectual disability, communication disorder), and those belonging to the SEN-B category (ADD/ADHD, autism, learning disability, behavioural and emotional disorders). Articles belonging to the SEN-C category were excluded as the problems behind the problems of SEN-C learners are different from those of the other two categories.

In the papers, various methods were used: eleven studies incorporated interviews, three used focus group conversations, eight conducted surveys, and four were based on case studies. Six papers introduced an intervention programme, and two presented a good practice. Some papers used mixed methods.

Resilience is measured in different ways. Most articles considered resilience an element of the personality and the basis of successful cooperation and parental involvement (16), while others regarded it as a consequence (12).

### 3.1. Research Focusing on the Attitudes of Children

Only a few studies could be found that focused on the perspective of children and young people. Flynn et al. (2012) investigated the parental involvement of seventh- and eighth-grader pupils in out-of-care, with and without disabilities [24]. According to their research, caregiver attitudes and behaviour are related to educational success. Caregiver involvement in several school activities and their aspirations were positively associated

with average marks and school performance. Involvement also had a positive impact on the GPA of eighth-grader pupils. Therefore, they strengthened the theory that caregivers are an important resource for improving educational outcomes.

In their study, Philips et al. (2000) presented the importance of Personalized Learning Plans (PLPs), which may also help engage students and parents in the planning process. A PLP is a useful tool for students with disabilities because it articulates specific educational needs. The study examined the practical application of PLPs through the experience of one case study. PLPs are better suited for each learner—with or without disabilities—since they each have their own learning style, needs and previous experiences [25].

Stainton and Bresser (1998) emphasised the positive contribution of children with intellectual disabilities to their families, such as giving joy and happiness, an expanded social network, community involvement, a family unit, personal growth and strength, increased tolerance, etc. The study was designed with semi-structured group interviews and two single-family interviews. The findings suggest that negative attitudes are deeply rooted in the assumption that having a disabled child is a tragedy. This research suggests that this assumption is false [26].

### 3.2. Research Focusing on the Attitudes of Parents

Even if the role of fathers is unquestionable, only a few studies focused on the impact of fathers' involvement, although it is well-known that it is positively associated with children's learning mental and emotional well-being. An early study by Linder et al. (1984) studied fathers' involvement with disabled children. The results suggest that the majority of fathers spend less than 20 h a week with their disabled children. The main responsibility for dealing with parenting issues and problems lies with mothers; however, four-fifths of fathers felt that the child's education and support was a shared responsibility of both parents and the school. The results show that fathers with children with disabilities are interested in their children's education, and 85.7% would like to participate in training and support programmes [27]. Therefore, it is important to actively involve fathers in early development programmes due to their needs and the positive impact of involvement on the child's development. Karisa et al.'s (2021) study used data from a broad qualitative case study of one special school from Kenya to better understand fathers' involvement in the formal education of children with intellectual disabilities. Fathers reported work commitments, lack of organisational and cultural support and concerns over the content of educational services as barriers to involvement [28]. The study highlights some key messages regarding fathers' involvement in their children's educational process: teachers' gender prejudices affect father involvement; the father's masculinity clashes with the child's disability, impacting father involvement; and coping mechanisms to threats to masculinity influence father involvement.

The comparative research by Lavan et al. (2019) considered resilience as a personality factor and highlighted different patterns in parents' overall stress levels and the coping styles of disabled and non-disabled children [29]. Their results point to a moderate positive relationship between parental involvement and the use of social support and emotional coping styles among parents of children with SEN, suggesting that building upon parental strengths and actively engaging them in a task-focused way may increase the parental involvement of both parents. At the same time, it is necessary to take into account the parents' sociodemographic characteristics, as the results suggest that parents with low educational attainment and a large number of children at home are less involved, presumably because it is difficult for them to find time to become engaged in school due to their daily responsibilities. They also tend to trust teachers' expertise (often because of their low self-confidence and lack of experience in education), leaving them to make decisions. However, the results related to the effect of SES are not consistent. Gedfie et al. (2020) also investigated parents' socioeconomic status regarding parental involvement in various activities, including parenting, communication activities, volunteering, home learning, community involvement and decision making [30]. The impact of socioeconomic status

was not significant. However, most parents of children with disabilities were involved prevalently in parenting activities rather than in other dimensions of parental involvement. Additionally, parents were not participating in communicating and learning at-home activities. Parents were not communicating with teachers regarding their children's daily progress frequently, which might have had an adverse effect on the academic and psychosocial development of children with disabilities. Furthermore, the level of involvement in volunteering, decision making and collaborating with the community was also below what was expected.

Stanley (2015) measured parental involvement and resilience among African American mothers of children with disabilities in rural areas using a strengths-based approach. According to the research results, parents agreed that advocacy begins early, but it usually does not take the same form, as it depends on the type of disability and the parents' personality. This also applies to individual and group advocacy, including other parents of children with and without disabilities. Open communication should be a necessity, but mothers reported it as a barrier since most teachers were driven by negative attitudes, using unknown terms without considering the parents' opinions. The mothers indicated a need for mutual respect and trust between parents and teachers to facilitate their advocacy efforts [31].

In their research, Buchner et al. (2014) also highlighted resilience as a key element in both children's academic progress and parental involvement, emphasising familial capitals forming resilience in hostile learning environments. As common characteristics, refusal and low levels of support can be identified as a shared experience of participants. The authors highlighted the role of social capital (e.g., one participant's mother founded a society to ensure that her child and others in similar shoes receive the appropriate treatment). Parents' coping strategies are passed on to their children. They concluded that resilience is not innate but part of a learning process that has developed in parents as a result of coping with difficulties and is passed on to their children. Social capital does not necessarily provide a straight path to resilience, but it does allow room for it to develop [32].

Stress seems to be a typical characteristic of families with children with special educational needs; therefore, parental involvement and effective support at home are crucial for families raising a child with learning disabilities. Pentyliuk (2002) suggested that a high level of stress is often present in families with children with disabilities, although families show good adaptation [33]. The study revealed that parents of children with learning disabilities needed a wide range of support and coping strategies in regard to the academic, social and emotional difficulties their children were experiencing in schools. Other studies also further demonstrated the crucial role of the personal strength of the parents. Miron (2010) focused on parents' views and perspectives of children with childhood apraxia of speech (CAS). The findings reinforce the importance of internal and external resources in promoting positive parent adaptation. The results highlight two recommendations for educators in their interactions with parents. First of all, most of the time, medical and educational communities are unhelpful and adversarial despite professionals' good intentions. Second, despite the overall negative experiences, professionals' positive attitude plays an important role in parents' adaptation experience [34].

Similar to Stanley et al., Casillas et al. (2000) focused on minority parents, however, they specified a target group with one type of disability [35]. The authors sought to understand the experiences of Latino and non-Latino White parents, emphasising the perspectives of fathers in raising a child with autism spectrum disorder (ASD). The authors used an exploratory qualitative research design to provide a platform for self-reflection for the parents. The study included six semi-structured interviews with eleven parents raising a child with ASD, and the study is considered unique because the fathers were also present. Two major findings emerged: firstly, there are similarities across all families' experiences of raising a child with ASD, and secondly, there are cultural differences between Latino and non-Latino participants. The study adds a cross-cultural perspective to raising a child

with ASD as well as to the inclusion of fathers' involvement in their child's education and family resilience.

The issue of school performance has always been one focus of research. Since most children with special needs also have to participate in education, it is essential to conduct research on this topic as well. Children with emotional and behavioural (EB) disorders also experience difficulties in academic achievement, social relationships and behaviours. Through qualitative analyses, Carlson et al. (2020) examined some positive and negative experiences among parents raising children with an EB disorder [36]. Involving parents in the Individualised Education Programme (IEP) strengthens proactive and positive communication and collaboration between the school staff and families. The study emphasised the importance of the cooperative parenting style, which influences behavioural outcomes for children with EB disorders, affecting academic achievement. Additionally, the issue of academic achievement has significantly come to the foreground due to the COVID-19 pandemic and the school closures. Singal et al. (2021) investigated this issue among children with special educational needs from a general perspective [37]. Besides the parents' background and the child's disability, they measured the schooling and learning of the child with disabilities during school closure and the impact of school closure on the child with disabilities and on the parents. The results of the interviews highlight that most parents (86%) reported they had had no contact with the school or the teachers during school closures; thus, school-related parental involvement faced multiple barriers. Therefore, parents expressed an urgent need for specialist support for children with disabilities even when schools were closed. Without professionals, they failed to support their children (e.g., they did not know sign language or read braille). Additionally, as a consequence, parents expressed their need for training to help their children when they are at home.

### 3.3. Research Focusing on the Attitudes of School Staff and Professionals

Rodriguez et al. (2014) focused on schools' efforts to facilitate parent involvement and parents' involvement with their child at school [38]. They highlighted that parent–school collaboration and communication varies, depending on the school's receptivity to parent input and the extent to which teachers actively solicited that input. Most parents were satisfied with schools' engagement efforts when they knew their children were making some progress. Additionally, parents' resilience is crucial as parents' comments suggested that schools became responsive only when they themselves took the initiative and became persistent or demanding in their requests. Furthermore, they concluded that more parent involvement does not necessarily mean better parent involvement. Therefore, the quality of support provided by schools is more important than the quantity of the options available.

Apodaca et al. (2015) consider parental involvement a mediator of academic achievement [39]. They investigated the topic from the perspectives of the student, their parents and teachers. The authors did not differentiate between the disorders of the students. Parents evaluated their involvement in their children's education along the five dimensions of parental involvement, i.e., parental expectations, parental communication, parental supervision, parental participation and general parental involvement. As an overall result, no significant relationships were found between any dimensions of parental involvement and grades in specific academic classes, which was true regarding gender. A positive relationship was found between parental expectations and overall student achievement. However, the study detected a negative correlation between the academic achievement of the student and the level of parental communication and general parental involvement. This result may show that the involvement of the parents does not always focus on improving the students' academic achievement, as this is not the only dimension in school life. However, having strong future expectations for the children may encourage parents to support the child's educational performance. Interestingly, the correlation between the teacher and parent ratings of parental involvement was not significant. Additionally, teacher ratings of parental involvement were not associated with student achievement.

In the research by Blackman and Mahon (2014), teachers distinguished two main groups of factors influencing parental involvement [40]. One group saw in-school factors, including teacher-dominated parent–teacher associations (PTA) and discontinuation of parent involvement initiatives and programmes as hindering factors, and teacher interest and initiative and active PTAs and parent conferences as positive factors. The other group referred to out-of-school factors, including collaboration, parents' coping, financial considerations and work commitments and scheduling of meetings. Overall, teachers reported a low level of parental involvement in both typologies. According to teachers, parents are not able to accept that their children with disabilities are different and therefore find it difficult to engage in the educational development process. Since coping and resilience are crucial in supporting children, it would be important to develop the personality of the parents and support them in coping with the difficulties.

Cultural and spiritual differences may also appear at the level of parental involvement. The study by Ravet and Mtik (2021) concluded that not only parental involvement but also children's school participation is a problem as some parents of children with disabilities do not wish to send their children to school [41]. On the one hand, this is the result of bullying as pupils' attitudes toward their disabled counterparts are often negative. On the other hand, it is due to a widespread belief, especially among rural communities, that disability can be attributed to spiritual causes and that children with disabilities cannot learn. Therefore, disability training would be necessary to share knowledge with parents and the local community to support the participation of students in education. This may also help their parents develop a resilient personality and visualise their possible positive role in their child's development.

Smith and Perlock (2002) described a case study through a case management model for speech-language pathologists working with school-age children with disabilities. School-based speech-language pathologists provide care coordination and consultation for students with multiple needs and their families [42]. The authors presented the necessary skills for effective case management through a case study describing a child and a family story. The more relevant skills needed for school-based speech-language pathologists are professionalism, teaming and cultural competencies. The Vermont Interdisciplinary Leadership for Health Professionals (VT-ILEHP) programme is a care coordination model whose goal is to teach these skills to professionals and educational teams and implement them for children with complex needs and their families. The study emphasised the importance of family-centred and culturally competent practice for children with multiple needs and challenges faced by professionals providing services that remain resilient.

### 3.4. Interventions

The intervention programmes can be separated based on the actors involved in the process and the main focus. Usually, we can find variations in terms of collaboration, but some intervention programmes build upon the idea of strengthening the personality and resilience of children with special educational needs and their parents. The article by Sochet et al. (2019) is an example of the latter [43]. Teenagers with autism spectrum disorder (ASD) have a greater risk of depression, which evolves into more compulsive, emotional and aggressive behaviour. Shochet et al.'s (2019) proof-of-concept study investigated the strength-focused parenting programme, analysing 15 parents' intervention experiences [43]. The authors highlighted the importance of parental involvement in the prevention approach to decrease the risk factors associated with obtaining the right support for children with ASD. The project entitled 2016 Adolescent Wellbeing was conducted across three urban schools in Brisbane, Australia. Parenting a young adolescent is often described by caregivers as an isolating, challenging and overwhelming experience.

The programmes differ according to the number of actors involved in the intervention. Some intervention programmes emphasise the collaboration between the school and the parents. Lendrum et al. (2015) summarised the main results of the Achievement for All (AfA) programme and its two strands, known as Assessing Pupil Progress (APP) and

structured conversations with parents (SCPs) [44]. The paper details the latter, emphasising the role of a clear framework for developing an open and ongoing dialogue with parents about their child's learning, leading to dynamic teacher–parent relationships and allowing for effective school–home partnerships based on strength and resilience. Two-thirds of the schools (66%) successfully completed at least one SC each year with the parents. Additionally, the decreasing proportion of schools who reported being unable to conduct any SCs with at least one parent reduced over time (38.8% of schools in the first year and 30.8% in the second year), showing that schools' efforts to engage with harder-to-reach parents were becoming more successful. The increasing level of the engagement of parents of children with specific learning difficulties and behavioural, emotional and social difficulties was also detectable due to the positive discussions between the school and parents instead of negative feedback on the behavioural issues and events.

In an effort to address the unique needs of students and families coping with attention-deficit/hyperactivity disorder (ADHD), Morris et al. (2019) developed the Family–School Success (FSS) programme to improve parental self-efficacy and child homework performance [45]. The FSS programme was delivered in a fee-for-service clinic setting, modified to be fully carried out through parents' group sessions. The participating families received homework tasks to encourage parents to implement and practise their parenting skills at each session. The study included a nine-session programme within 18 cohorts of FSS delivered by three licensed psychologists. The study showed significant improvement in parental self-efficacy, child homework performance and reduction in child impairment throughout the programme. Parents practising FSS skills assisted their child's education more effectively and reported that their children, who had ADHD, were more productive with their homework. In their study, Power et al. (2009) also investigated the Family–School Success (FSS) programme, hwich engages families and schools coping with attention-deficit/hyperactivity disorder (ADHD) through the Coping with ADHD through Relationships and Education (CARE) programme [46]. The study included 93 cases; 45 participated in FSS, and 48 were in the CARE programme. FSS was designed to improve family involvement in education and build a family–school partnership, with 12-session interventions actively involving the children while the parents were in group sessions. The 12-session CARE programme was similarly designed to support groups of parents coping with ADHD, but its purpose was to control for the non-specific effects of the intervention. The findings of this study indicate that the FSS programme involved teachers in interventions much more than the CARE programme did. Involving teachers in programmes such as FSS provides more information about the family–school partnership.

In their study, Fabiano et al. (2021) highlighted the major need for male caregivers' engagement within educational settings. Children with attention-deficit/hyperactivity disorder (ADHD) show significant academic underachievement, which may be attenuated by the benefits of parental involvement [47]. The study presents The Coaching Our Acting Out Children: Heightening Essential Skills (COACHES) programme designed for elementary school settings. Male caregivers involved in the COACHES programme were encouraged to develop a growth mindset and become more resilient by praising their children's efforts.

We can also see an example of involving a mentor in the process from the side of the school. In their study, Swoszowski and Rollins (2019) presented the creation of the check-in/check-out (CICO) and check-in/check-up/check-out (CICUCO) interaction programmes in a case study [48]. Both intervention programmes are observation- and point-based programmes in which the student is assessed on arrival at school and departure from school. The elements of the CICO programme are: (a) check-in with the mentor, (b) point feedback per hour/period from the teacher, (c) check-out with the mentor, (d) home meeting/signing and (e) turning in the point sheet at school the following morning (p. 1). In comparison, CICUCO is extended with a monitoring component based on a joint discussion between the student and the mentor to discuss triggers and ways of problem solving to better respond to frustrating or difficult situations, resulting in feedback from the mentor at the end. Both programmes have a home-based component that relies on

parental involvement (parent looks at form, signs it and puts it in folder/bag, and child returns it to school the next day). It is particularly effective for pupils with behavioural problems. They are monitored through the programme to observe the reasons behind the problematic behaviour. It has been tested in both mainstream and special school settings and has shown positive results in both.

Some intervention programmes extend the idea of parent–school collaboration with the involvement of professionals, thus creating a multidisciplinary team collaboration between the actors. Santamaria Graff et al. (2021) emphasised the operation of the Family as Faculty (FAF) programme, which was developed to improve communication between doctors, health professionals and families of children with disabilities [49]. It builds on the principle that families can teach health professionals how to listen, understand and support individual family needs better. In the programme, parents and family members of children with disabilities provide important insider information to (higher education) students becoming health professionals through structured presentations [50]. For the 22 special education students who participated in the research, the programme reinforced the view that parents are experts, something which can help to slowly and positively transform students' deficit-based assumptions about families. Therefore, the FAF programme strongly supports resilience through developing resilient communication between the actors.

Since children with disabilities are considered pupils at risk of school failure, Slowik et al. (2021), as a conclusion of the results of the 'Pathways to Inclusion' project conducted in the Czech Republic, emphasised the role of cooperation between the school, family and social services to support pupils at risk of school failure [51]. On the one hand, they suggested stimulating support for pupils in families and providing parents with workshops to improve parents' attitudes to school and their children's education. On the other hand, they proposed seminars for parents and workshops for teachers to help them to meet the problems of the child and the family. As a good practice, they introduced team-based support of students at risk of school failure through case conferences, in order to find out about special cases, share experiences and strengthen cooperation between the actors. A similar type of collaboration between the actors can be seen in the paper by Timothy and Agbenyega (2019) too [52]. It is well-known that Individualised Education Plans (IEPs) are considered an element of the development of children with special needs, as highlighted by several research results and good practices. However, in their study, the authors refer to a paradigm shift from providing isolated services for children with SEN to a more collaborative and team-orientated approach to sharing strategies. They emphasised the positive impact of developing IEP for children through involving the parents. An IEP is created based on the child's characteristics, using their strengths and reducing their weaknesses. Parents have relevant knowledge; thus, involving them in the process may lead to a more accurate IEP and may increase the engagement and active participation of the parents in the child's educational development. Therefore, a collaboration between school leaders, teachers and parents with common goals may effectively support students through IEPs.

## 4. Conclusions

According to previous research results, parental involvement is considered an empowering factor in improving academic and non-academic achievement of school-aged children. It is also of paramount importance to highlight its necessity in the case of families with children with special educational needs. Education and health policy is also concerned with this issue. Education is an obligatory part of children's life (except for those with serious disorders); however, their schoolwork and efficacy are often hindered by their disorders. Therefore, health policy, as the responsible field for supporting these families with different treatments including drugs and (psycho)therapies, is also concerned with supporting the healing process of the child and the cooperation between families with children with special needs and health and school actors.

In conclusion, a huge variety in the elaboration of the research focus can be seen. Concerning resilience, two types of papers have been identified. The majority of papers consider resilience as a trait and element of parental involvement which can support children's academic and non-academic achievement [25,26,28,29,31–35,38–40,48,49,51]. Intrapersonal factors and psychological characteristics can support positive attitudes developed by the parents and may help the parents and the family cope with the special situation caused by the child's disability. The other type of article refers to the other side of the argument, i.e., the lack of coping flexibility and resilience of the parents and families [24,27,30,36,37,41,43–47,51]. In this case, resilience is considered a consequence, usually as a result of intervention or collaboration between the actors.

From a methodological perspective, most researchers prefer qualitative research methods, typically due to the specificity of the target group and the need for deep analysis of the topic and related phenomena. Interviews make it possible to meet the perception of the different actors in depth. It is clear, however, that most research focuses on the attitudes and experiences of parents [27–37]. Investigating the parents' perspectives is of paramount importance when investigating parental involvement. Another significant number of papers reflected on the experience of teachers [38–41] or professionals [42], which is also an essential element in the topic. However, there is a lack of research investigating the children's experience [24–26]. It would also be necessary to meet the perceptions and needs of children and young people since they are the ones experiencing the disability. As most types of disability are not associated with intellectual disability, there is no problem of interpretation that could hinder the exploration of the attitudes and needs of children and young people. As development is aimed directly or indirectly at them, it is essential that their specificities and experiences are taken into account. This area of research should be emphasised in the future.

Interventions are also crucial in the issue of parental involvement. Most programmes focus on a strong and mutual collaboration between the school and family [43–47]. The communication between parents and teachers is of paramount importance based on the interview results, and it is visualised in the intervention programmes as well (AfA) [44]. In some cases, it is mediated by school mentors (CICO and CUCICO) [48]. Some programmes are adapted from others [45–47], and others are supported by external professionals, e.g., health professionals [49] or social services (Pathways to Inclusion) [51]. Nowadays, multi-disciplinary collaborations are gaining ground, as they approach the problem from multiple perspectives and aim to tackle the problem with a multidisciplinary professional tool [53], including medical providers, social workers, occupational therapists, speech and language pathologists, nurses, clinical psychologists, behaviour interventionists and the core family–school partnership team (e.g., children, families, schools and school psychology faculty and graduate students), depending on the type and severity of the disability [54].

When focusing on parental involvement, we also have to take the school's engagement into account. Although some research indicates that parents are less involved in school-related activities, especially those with a low socioeconomic status, some other researchers highlight the need for parents with a low SES to create contact between the school and families [29,30]. However, they also lack knowledge and information and need another type of communication. Therefore, the flexibility and resilience of the school staff must be emphasised too, since they have to adapt to the situation and try to find an adaptive way to communicate and cooperate with families. The intervention programmes usually integrate communication development and facilitation elements, which can significantly decrease miscommunication-related problems [44,49]. The duality of resilience can also be experienced in the case of the intervention programmes since some build upon the existing personality and resilience of the parents, while others aim to develop it as a result of the programme. The latter is critically important for parents who are hard to reach and to involve in the student's academic life. It is known that parents with a low socioeconomic status are considered vulnerable in terms of parental involvement due to

a lack of knowledge or interest and the increased number of tasks they must complete, especially if they have a large family.

Our study suggests that working with parents with special educational needs involves three main categories: communication, involvement and support [55]. This literature review highlights that parents with children with special educational needs have a lot to offer to professionals, which can improve the outcomes of the educational process. SEN schools hold regular sessions, including clinics and therapy-informed meetings, to support institutions in examining parent's attitudes, and they offer a wide range of additional support, such as occupational therapy, physiotherapy, speech and language therapy, play and music therapy, animal-assisted therapy, etc. However, the supports mentioned above were often just briefly mentioned in the selected papers.

Overall, resilience is an element of parental involvement, either as a personality trait or a result. Its presence is necessary for the successful development of the children's academic and non-academic achievement. In general practice, a strong collaboration between parents, teachers, school psychologists and special education teachers can significantly contribute to the child's psychological, emotional and academic development. The findings of our study show that parents have an important role in their children's education; therefore, it is essential that all professionals actively seek collaboration with parents, since for the development of a reliable, resilient family–school relationship, parents need to feel included. From the perspective of the parent–school collaboration, developing a resilient communication style is one of the most important cornerstones of creating long-lasting and persistent parental involvement.

**Author Contributions:** Conceptualization, A.H. and G.P.; methodology, K.B.; validation, K.E.K., B.D. and A.H.; formal analysis, K.E.K.; investigation, K.E.K. and B.D.; writing—original draft preparation, K.E.K., B.D., A.H. and K.B.; writing—review and editing, G.P. and K.B.; visualization, K.B.; supervision, G.P.; funding acquisition, G.P. All authors have read and agreed to the published version of the manuscript.

**Funding:** The research on which this paper is based has been implemented by the MTA-DE-Parent-Teacher Cooperation Research Group and with the support provided by the Research Programme for Public Education Development of the Hungarian Academy of Sciences. This paper was supported by the János Bolyai Research Scholarship of the Hungarian Academy of Sciences.

**Institutional Review Board Statement:** This research was conducted in accordance with the Declaration of Helsinki. The ethical committee of the University of Debrecen approved this study. The protocol code is: 1/2022. The research is conducted ethically, the results are reported honestly, the submitted work is original and not (self-)plagiarised, and authorship reflects the individuals' contributions.

**Informed Consent Statement:** Informed consent was not required for the study.

**Data Availability Statement:** Data are available only on request.

**Conflicts of Interest:** The authors declare no conflict of interest.

## Appendix A  Databases Involved in Data Collection by EBSCO Discovery Service

Accucoms—COVID-19 resources,
ACM Digital Library
Arts & Humanities (Proquest)
Bibliotheca Corviniana Digitalis
Biological Abstracts 2000–2004
Biomedical & Life Sciences Collection
*BMJ* Journals
Business Source Premier
CAB Abstracts
Cambridge Journals
ChemSpider
CNKI
Cochrane

COMPASS
Congress.gov
De Gruyter Journals
Directory of Open Access Journals (DOAJ)
Ebook (Springer)
Ebook Collection (Ebsco)
EbookCentral (Proquest)
EBSCOHost
Elsevier
Elsevier—SciVal
ELTE Reader
Emerald
EMIS University— Central and Southeast Europe
EndNote
ERIC
European Parliament Legislative Observatory
EUR-Lex
Europeana Collections
EUROSTAT
FSTA (Food Science and Technology Abstracts)
GALE Literary Sources (GLS)
Gale Reference Complete
Global Health and Human Rights Database
Grove Music Online
HUMANUS
HUNGARICANA
IJOTEN
Impact Factor (Journal Citation Reports)
InCites
International Human Rights Network
Internet Archive
JSTOR
MATARKA
MathSciNet
MathSciNet (EBSCOhost)
MEDLINE (EBSCOhost)
MEDLINE (PubMed)
Medscape
Nature Journals
NEJM Group—COVID-19 resources
*Nutrition and Food Sciences*
Oxford Handbooks Online (OHO)—Criminology and Criminal Justice
Oxford Handbooks Online (OHO)—Law
Oxford Scholarship Online (Law Collection)
Oxford University Press (OUP) Journals
Project Gutenberg
ProQuest—One Academic
PubMed
PubMob
RefWorks
SAGE Journals
Science Direct
*Science Magazine*
SciFinder
Scifinder-n
SCImago Journal and Country Rank (SJR)
SciTech (Proquest)
SCOPUS
SHERPA/RoMEO

SpringerLink
STADAT
Statista
SzocioWeb
Taylor and Francis Online Library
The Historical Map Portal
United Nations Treaty Collection
UpToDate Advanced
Web of Science
Wiley Online Library
World Biographical Information System
zbMATH
International Directory of Music Resources

**Special Journals:**

*Education Sciences*, *European Journal of Special Needs Education*, *International Journal of Inclusive Education*, *Teacher Education and Special Education*, *Disability and Society*, *Exceptional Children*, *Journal of Research in Special Educational Needs*, *Teaching Exceptional Children*, *International Journal of Special Education*, *Support for Learning*, *Exceptionality Education International*

**Appendix B  Inclusion Criteria and the Scheme for Annotations**

Study information:

- Author(s)
- Study title
- Publication year
- Journal

Main topic:
Type of SEN:
Resilience:
Population:
Prevention/intervention program
Study design:

- Study type (e.g., survey, interview, longitudinal study)
- Objectives/purpose

Research focusing on the attitudes:

- Parents
- Children
- School staff and professionals

Parental involvement:
Results:

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
