# Peer review of "Is Resilience a Trait or a Result of Parental Involvement? The Results of a Systematic Literature Review"

_education, doi:10.3390/educsci12060372_

Round 1

Reviewer 1 Report

The article addresses an important topic but is confusing. Much of the text does not address the stated focus of the article and includes sweeping statements not based on evidence. Many of the in-text citations do not match the statements they are meant to support. Key elements of the literature on parental involvement with parents of children with disabilities are not included. There are many errors in the reference list at the end, Some of the English is confusing.

Author Response

We would like to thank the reviewer for her/his work. Our point-by-point responses are see below, please. For the corrected text please see the attachment.

The article addresses an important topic but is confusing. Much of the text does not address the stated focus of the article and includes sweeping statements not based on evidence. Many of the in-text citations do not match the statements they are meant to support.

The aim of the paper was to cover the patterns of previous articles related to parental involvement focusing on children with special education needs in the light of resilience. The structure of the analysis follows the pattern we could detect. The best categorization could have been based on the actors of the papers. However, since each paper contained resilience as well, it was a basic characteristic of each article. Since resilience was always the focus of the analyses (examining its level or status), we could categorize its presence concerning its interpretation (namely as a trait or a result) which was detailed in the Results and Conclusions sections.

Key elements of the literature on parental involvement with parents of children with disabilities are not included. The collection of literature was based on the described methodology. We used the search engine of the EBSCO Discovery Service. As we wrote, Books/Book Chapters, Dissertations, Newspaper Articles were excluded. Because of that basic works can be left out. Our systematic review study focuses on that literature which we on this way can rich. We carefully revised the results, so we are sure that the key elements, which you are missing, are not in our result list. So they are not available with this search method.
  There are many errors in the reference list at the end
The reference list is created automatically using Zotero. Thank you for drawing our attention to the errors. We have fixed the inaccuracies.

Reviewer 2 Report

This is a quite useful systematic review which can make a significant contribution in the literature related to resilience and children with special educational needs. Authors make crucial interpretation of the notion of resilience both as a personality trait or consequence and they highlight its major  importance  for the successful development of the children both in academic but also in their social-emotional life. Authors also highlight  the beneficial aspect of collaboration between parents, teachers and professionals and  their vital contribution to offer elements of resilience strategies and finally  'cure' the traumatic elements of children's with special educational needs and their families in their daily lives. 

Author Response

We would like to thank the reviewer for her/his work. We also think that resilience and the partnership between parents and teachers are very important in the life of students with special educational needs.

Reviewer 3 Report

Thank you for the opportunity to review the manuscript entitled, “Is Resilience a Trait or Result in Parental Involvement? The Results of a Systematic Literature Review”, submitted to Educational Sciences. The manuscript addresses a topic that should be of interest to readers and supports the call of the Special Issue: Building Resilience of Children and Youth with Disabilities: New Perspectives. Below are suggestions that the authors might find helpful in revising this manuscript and refining their research methods.

Introduction:

            The authors present a strong introduction of why home/school collaboration matters, especially for children and youth with disabilities. The authors also give context to how disability is defined globally and differences in how disability is conceptualized across various cultures. While this context is important, its relevance to the literature review is not explicitly stated. The authors provide a table detailing how disability is categorized according to “OCED” (though not defined), though it is unclear why this categorization is imperative to understanding this literature review. It would be helpful if the authors included which specific disability categories fits each category (e.g., SEN-A, SEN-B) and how the categorization influenced and/or explains the major findings of their literature review.

Method:

            I have a few major concerns with how the methods are reported and would need clarification on these matters in order to consider this manuscript for publication. First, the authors report finding 442 studies in their database search, with 137 being duplicates. They also report finding 183 “unique results” from a hand search of 11 special education journals. I believe the hand search results are extremely inflated if the search was correctly performed. The keywords used for the search include special education and disabilities, leading me to believe the 183 results from the hand search should have been caught during the database search. It would be helpful if the authors explained why their hand search had an uncharacteristically high yield.

            My next concern with the authors’ methods is lack of interobserver agreement (IOA) reported for the screening process. While it is mentioned that a second rater reviewed 25% of the titles and abstracts for Stage 1, no IOA is reported for Stage 2. This is highly inconsistent with standard screening practices for systematic reviews, which usually include a second rater for at least 30% of the data but also the percentage of agreement is reported to ensure the inclusion criteria was systematically implemented.

            My last concern is with the inclusion criteria and included studies, which is related to the previous concern of IOA. The authors report their inclusion criteria to be “(1) reported original, empirical research published in a peer-review journal…”. However, I am actually one of the authors of their included “studies” and my article does not fit this inclusion criteria. My article is a practitioner-based article, reporting how to incorporate an intervention into a classroom. The authors note in Table 2 that our article includes “intervention” but it does not. It does not report “original, empirical research”; it’s a research-to-practice paper. However, there are multiple articles reporting original, empirical results for check-in/check-out or check-in/check-up/check-out that are not included in this review, but are published in special education journals. This inconsistency reinforces my concerns with how the search was performed. With these concerns, I was not able to effectively review their results and discussion.

            In conclusion, I believe the subject of this literature review is needed and important for the field and the authors begin to present a convincing argument for why home/school collaboration should be prioritized for families with disabilities. However, the authors’ methods need to be more systematic to effective conduct this review. As it is written, I do not recommend this manuscript for publication, but hope the authors will incorporate this feedback into a future, revised search for this topic.

Author Response

We would like to thank the reviewer for her/his work. Our point-by-point responses are see below, please. For the corrected text please see the attachment.

The authors provide a table detailing how disability is categorized according to “OCED” (though not defined), though it is unclear why this categorization is imperative to understanding this literature review. It would be helpful if the authors included which specific disability categories fits each category (e.g., SEN-A, SEN-B) and how the categorization influenced and/or explains the major findings of their literature review.

Response: This classification is particularly important in the analysis because it has allowed us to include in the search keywords that also cover those types of special educational needs that result from disabilities or difficulties, even internationally. In the analysis, therefore, this will not form the basis of a comparison. However, during the screening process, we attached great importance to it, excluding research on special educational needs in the non-SEN-A and non-SEN-B categories. Also, the number of papers did not allow us the in-depth analysis of the papers alongside the SEN-A and SEN-B categories. Therefore, we indicated the focus of each research to show a general image about the papers but did not use them as further analysing categories.

They also report finding 183 “unique results” from a hand search of 11 special education journals. I believe the hand search results are extremely inflated if the search was correctly performed. The keywords used for the search include special education and disabilities, leading me to believe the 183 results from the hand search should have been caught during the database search. It would be helpful if the authors explained why their hand search had an uncharacteristically high yield.

Response: Thank you, we've clarified the description of our search methodology. We additionally searched in 11 special Journals. A search on SCImago using selection criteria: "social sciences", "education", "all regions", "journals", "2020"; ten journals were identified with potentially relevant topics regarding our systematic review. The 11th Journal was the Education Science. After an initial screening of the keywords yielded a total of  183 unique studies (see Appendix A). For screening, we used the own search box of the Journals and used the same keywords.

My next concern with the authors’ methods is lack of interobserver agreement (IOA) reported for the screening process. While it is mentioned that a second rater reviewed 25% of the titles and abstracts for Stage 1, no IOA is reported for Stage 2. This is highly inconsistent with standard screening practices for systematic reviews, which usually include a second rater for at least 30% of the data but also the percentage of agreement is reported to ensure the inclusion criteria was systematically implemented.

Response: Stage 2, screening of full texts: We worked in screening pairs (KK with DB and HA with BK). The set of full texts has been split into two parts. Each screening pair screened one of these parts. Within a screening pair, all the papers that were assigned to the screening pair were screened by both members. The results of the two independent screening within a pair were compared, paper by paper, and if there was a difference in the screening result of a given paper, a review author from the other pair was involved in forming a quorum and making a decision.

My last concern is with the inclusion criteria and included studies, which is related to the previous concern of IOA. The authors report their inclusion criteria to be “(1) reported original, empirical research published in a peer-review journal…”. However, I am actually one of the authors of their included “studies” and my article does not fit this inclusion criteria. My article is a practitioner-based article, reporting how to incorporate an intervention into a classroom. The authors note in Table 2 that our article includes “intervention” but it does not. It does not report “original, empirical research”; it’s a research-to-practice paper. However, there are multiple articles reporting original, empirical results for check-in/check-out or check-in/check-up/check-out that are not included in this review, but are published in special education journals. This inconsistency reinforces my concerns with how the search was performed. With these concerns, I was not able to effectively review their results and discussion.

Response: We agree with the reviewer concerning the type of the article since it introduces two programs in detail. However, due to the introduction of the case of Antony, we considered the paper as an empirical study as it is similar to a case study. The case study method is a qualitative data gathering method in an empirical research study. It involves sifting through and analyzing relevant cases and real-life experiences about the research subject or research variables in order to discover in-depth information that can serve as empirical data.  Therefore, we considered it important to involve this paper in the review process.

Round 2

Reviewer 1 Report

Article is not up to standard required for publication

Reviewer 3 Report

Dear authors, 

Thank you for addressing previous feedback provided and clarifying your methods accordingly. My primary concern was about your screening process and results, which you addressed. You expanded the description of your inclusion criteria to include research-based reports/research-to-practice articles, which explains how some of your article were accepted. I think this practice makes your review very broad in nature, but acceptable nonetheless. Thank you for your meaningful work to this field and your dedication to this review.